# Equivariant Geodesic Networks: Geometry Preserving Learning on Riemannian Manifolds

## Abstract

Many high-dimensional data modalities—including covariance descriptors, diffusion tensors, and kernel matrices—naturally reside on Riemannian manifolds such as the space of Symmetric Positive Definite (SPD) matrices. However, conventional deep neural networks often fail to respect the intrinsic geometry of such data, leading to suboptimal representations and generalization. We introduce Equivariant Geodesic Networks (EGN), a novel architecture designed to operate directly on Riemannian manifolds while preserving key geometric properties. EGN incorporates manifold-consistent operations, including equivariant mappings, adaptive geometric bias, and structured low-rank updates that respect the underlying topology. Unlike existing methods that either flatten or project SPD data into Euclidean space, EGN directly learns on the manifold, preserving geometric consistency throughout. We provide theoretical analysis of the manifold-preserving properties of our layers and demonstrate significant empirical gains on tasks involving SPD-valued data, such as EEG-based emotion recognition and imagined speech classification. EGN outperforms existing Euclidean and pseudo-manifold baselines, offering a principled approach to end-to-end learning on Riemannian data manifolds.

## 1 Introduction

In deep learning, data is typically represented as vectors in Euclidean space $\mathbb{R}^n$. However, high-dimensional data types like covariance descriptors, diffusion tensors, and EEG connectivity matrices naturally reside on curved spaces, specifically forming Symmetric Positive Definite (SPD) matrices on Riemannian manifolds (Barachant et al., 2012). Unlike flat Euclidean spaces, these manifolds impose structural constraints, making direct learning essential to preserve geometric consistency (Fletcher et al., 2004). Projecting SPD matrices to Euclidean space can distort relationships (Arsigny et al., 2006). While leveraging the manifold's intrinsic geometry ensures positive definiteness and symmetry, leading to more accurate modeling in applications like EEG-based emotion recognition and diffusion tensor imaging (Congedo et al., 2017).

A Symmetric Positive Definite (SPD) matrix is a symmetric square matrix that satisfies positive definiteness for any non-zero vector $\mathbf{v} \in \mathbb{R}^n$(Arsigny et al., 2006). These matrices inherently preserve geometry through positive eigenvalues and symmetry, capturing spatial and structural relationships (Han et al., 2024).

Conventional deep learning methods, rooted in Euclidean geometry, face inherent challenges when applied to Symmetric Positive Definite (SPD) matrices, $\mathcal{S}_{-}++^n$, which naturally lie on a Riemannian manifold (Pennec et al., 2006). Linear operations and distance metrics, when replaced with Euclidean approximations, fail to preserve geometry, breaking positive definiteness and symmetry. As a result, applying standard neural network layers to SPD data often distorts embeddings, failing to capture the manifold's curvature. Proper handling of SPD data requires Riemannian geometry to maintain the manifold structure and preserve essential geometric properties, ensuring valid representations (Tang et al., 2021).

In this study, we introduce the Equivariant Geodesic Network (EGN), a novel framework for learning directly on Riemannian manifolds, specifically targeting the space of SPD matrices, denoted

as $\mathcal{S}_{++}^n$. Unlike traditional methods that flatten manifold-valued data into Euclidean space, EGN preserves the geometric structure throughout the learning process. The core innovation of EGN lies in maintaining both equivariance and geodesic consistency, where equivariance ensures consistent representations under geometric transformations, and geodesic consistency accurately measures distances on the manifold using the affine-invariant Riemannian metric (AIRM). This approach inherently respects the curvature of SPD data, mitigating distortions from flattening. Additionally, EGN incorporates a data-driven geometric bias mechanism to adapt to underlying distributions, enhancing its ability to model complex manifold relationships. We evaluate EGN on emotion recognition, imagined speech classification, and Psychiatric disorder using different EEG datasets.

While previous methods have incorporated individual Riemannian components such as log-Euclidean layers, manifold pooling, or SPD-preserving transformations, these have mostly been limited to partial or shallow integrations. In contrast, EGN offers the first fully end-to-end geometry-preserving deep network that coherently integrates equivariant mappings, learnable geometric bias, true Riemannian pooling, and geodesic attention-based classification, all trained via manifold-specific backpropagation. This level of integration ensures geometric consistency across all layers, which we demonstrate leads to significant empirical gains.

Our main contributions are:

    i. We propose the first fully-integrated Riemannian deep network (EGN) combining equivariant mapping, geometric bias, and geodesic attention in an end-to-end SPD-preserving architecture.

    ii. We introduce the Geometric Bias Block for learning adaptive, manifold-consistent bias transformations.

    iii. We design a geodesic attention classifier using affine-invariant distances and prototype matching.

    iv. We develop a Riemannian backpropagation strategy via gradient lifting to ensure geometry-aware optimization.

    v. We develop a Riemannian-specific backpropagation method for efficient training.

    vi. While prior work explores isolated SPD components, our novelty lies in unifying them into a scalable, coherent framework with theoretical guarantees.

## 2 BACKGROUND

In this section, we describe the fundamental concepts and mathematical foundations essential for understanding the proposed Equivariant Geodesic Network (EGN).

### 2.1 RIEMANNIAN GEOMETRY

Riemannian geometry provides the foundational framework for analyzing curved manifolds, extending classical Euclidean concepts to non-Euclidean spaces. A Riemannian manifold $(\mathcal{M}, g)$ is a differentiable manifold $\mathcal{M}$ endowed with a Riemannian metric $g_p$ at each point $p \in \mathcal{M}$, which defines an inner product on the tangent space $T_p\mathcal{M}$. Formally, the metric is given by $g_p(\mathbf{v}, \mathbf{w}) = \langle \mathbf{v}, \mathbf{w} \rangle_p$, which induces the norm $\|\mathbf{v}\|_p = \sqrt{g_p(\mathbf{v}, \mathbf{v})}$. The length of a smooth curve $\gamma : [0, 1] \to \mathcal{M}$ connecting $p$ and $q$ is $L(\gamma) = \int_0^1 \|\dot{\gamma}(t)\|_{\gamma(t)} dt$, and the geodesic distance $d(p, q)$ is the infimum of $L(\gamma)$ over all such curves (Becigneul & Ganea, 2019).

### 2.2 SPD MANIFOLD

The set of symmetric positive definite matrices of order $n$, denoted as $S_{++}^n$, forms a Riemannian manifold. A matrix $\mathbf{X} \in S_{++}^n$ satisfies:

1. **Symmetry:** $\mathbf{X} = \mathbf{X}^T$    2. **Positive Definiteness:** $\mathbf{v}^T \mathbf{X} \mathbf{v} > 0 \quad \forall \mathbf{v} \in \mathbb{R}^n \setminus \{0\}$

The space $S_{++}^n$ is equipped with a Riemannian metric defined as:

$$g_{\mathbf{X}}(\delta\mathbf{X}_1, \delta\mathbf{X}_2) = \mathrm{Tr}\left((\mathbf{X}^{-1}\delta\mathbf{X}_1)(\mathbf{X}^{-1}\delta\mathbf{X}_2)\right). \tag{1}$$

This metric respects the manifold structure, as opposed to the Euclidean inner product which disregards symmetry and positive definiteness (Han et al., 2021). The tangent space at point $\mathbf{X}$ is:

$$T_{\mathbf{X}}S_{++}^n = \{\delta\mathbf{X} \mid \delta\mathbf{X} = \delta\mathbf{X}^T\}. \tag{2}$$

## 2.3 EQUIVARIANT OPERATION

An operation $\phi$ on SPD matrices is said to be equivariant under a transformation group $G$ if:

$$\phi(g \cdot \mathbf{X}) = g \cdot \phi(\mathbf{X}) \quad \forall\, g \in G, \mathbf{X} \in S_{++}^n. \tag{3}$$

In the context of our architecture, equivariance ensures that transformations applied to input data are reflected in the output without distortion (Zhouyin et al., 2025; Lin et al., 2024). This property is critical for manifold-based learning as it maintains consistency when the data undergoes transformations that preserve the manifold structure. For instance, if $g$ represents a congruence transformation, the equivariance operation should satisfy:

$$\phi(\mathbf{P}\mathbf{X}\mathbf{P}^T) = \mathbf{P}\phi(\mathbf{X})\mathbf{P}^T. \tag{4}$$

## 2.4 AFFINE-INVARIANT RIEMANNIAN METRIC AND GEODESIC DISTANCE

The Affine-Invariant Riemannian Metric (AIRM) defines a natural geometry over the space of symmetric positive definite (SPD) matrices. Given two SPD matrices $\mathbf{X}, \mathbf{Y} \in \mathcal{S}_{++}^n$, the AIRM-induced geodesic distance is defined as:

$$\delta_{\mathrm{AIRM}}(\mathbf{X}, \mathbf{Y}) = \|\log(\mathbf{X}^{-1/2}\mathbf{Y}\mathbf{X}^{-1/2})\|_F, \tag{5}$$

where $\log(\cdot)$ denotes the matrix logarithm, and $\|\cdot\|_F$ is the Frobenius norm. This distance captures the shortest path between $\mathbf{X}$ and $\mathbf{Y}$ on the Riemannian manifold, respecting its curvature and affine-invariance. In EGN, we exploit this structure-preserving distance to ensure that all operations align with the intrinsic geometry of SPD-valued data.

The AIRM is invariant under affine transformations:

$$\delta(\mathbf{P}\mathbf{X}\mathbf{P}^T, \mathbf{P}\mathbf{Y}\mathbf{P}^T) = \delta(\mathbf{X}, \mathbf{Y}) \quad \forall\, \mathbf{P} \in \mathrm{GL}(n). \tag{6}$$

This property is essential when the data represents covariance matrices or other structured forms where affine transformations are natural.

## 2.5 RIEMANNIAN MEAN

The Riemannian mean of a set of SPD matrices $\{\mathbf{X}_i\}_{i=1}^N$ is the matrix $\mathbf{G}$ that minimizes the sum of squared AIRM distances (Sra, 2012):

$$\mathbf{G} = \arg\min_{\mathbf{Y} \in S_{++}^n} \sum_{i=1}^N \delta_{\mathrm{AIRM}}^2(\mathbf{X}_i, \mathbf{Y}). \tag{7}$$

An iterative formula for computing the mean is given by:

$$\mathbf{G}_{k+1} = \mathbf{G}_k \exp\left(\frac{1}{N}\sum_{i=1}^N \log(\mathbf{G}_k^{-1/2}\mathbf{X}_i\mathbf{G}_k^{-1/2})\right). \tag{8}$$

This formulation ensures that the resulting mean remains on the SPD manifold, preserving its intrinsic geometry. We compute the Riemannian (or Karcher) mean using a fixed-point update, following the formulation in (Pennec, 2006).

# 3 METHODOLOGY

In this section, we provide technical details behind Equivariant Geodesic Networks.

## 3.1 ARCHITECTURE OF EQUIVARIANT GEODESIC NETWORK (EGN)

The Equivariant Geodesic Network (EGN) is designed as a fully end-to-end architecture that operates directly on Riemannian manifolds, specifically on the space of Symmetric Positive Definite (SPD) matrices, denoted as $\mathcal{S}_{++}^d$. The model leverages a sequence of manifold-preserving transformations, beginning with an SPD-forming operation $\mathcal{F} : \mathbb{R}^{d \times d} \to \mathcal{S}_{++}^d$, which ensures that the input matrices maintain their positive definiteness. This is followed by an equivariant bilinear transformation $\mathcal{B} : \mathcal{S}_{++}^d \to \mathcal{S}_{++}^m$, which maps the input to a higher-dimensional SPD space while preserving geometric properties. Subsequently, a manifold-aware activation function $\mathcal{A} : \mathcal{S}_{++}^m \to \mathcal{S}_{++}^m$ is applied, incorporating nonlinear geometric transformations that respect SPD constraints. To introduce adaptive bias that respects the intrinsic structure, we employ a geometric bias operation $\mathcal{G} : \mathcal{S}_{++}^m \to \mathcal{S}_{++}^m$, followed by Riemannian mean aggregation $\mathcal{M} : (\mathcal{S}_{++}^m)^k \to \mathcal{S}_{++}^m$ to capture average geometric features. Attention mechanisms are then applied via a geodesic distance computation $\mathcal{D} : \mathcal{S}_{++}^m \times \mathcal{S}_{++}^m \to \mathbb{R}^c$, which measures distances between data points and prototypes. Finally, an attention-based prediction layer $\mathcal{P} : \mathbb{R}^c \to \Delta^{c-1}$ generates classification probabilities by leveraging manifold-consistent softmax operations. The sequence of operations within the Equivariant Geodesic Network (EGN) can be mathematically formulated as follows:

$$X \xrightarrow{\mathcal{F}} \Sigma \xrightarrow{\mathcal{B}} \Sigma' \xrightarrow{\mathcal{A}} \Sigma'' \xrightarrow{\mathcal{G}} \Sigma''' \xrightarrow{\mathcal{M}} \mu \xrightarrow{\mathcal{D}} \alpha \xrightarrow{\mathcal{P}} \hat{y}$$

$X$ denotes the input data. $\Sigma$ is the SPD-formatted matrix obtained after transformation $\mathcal{F}$. $\Sigma'$ is the equivariant mapping output via $\mathcal{B}$, while $\Sigma''$ is the activated matrix post non-linear activation $\mathcal{A}$. $\Sigma'''$ represents the geometric bias-adjusted output via $\mathcal{G}$. $\mu$ is the Riemannian mean after pooling operation $\mathcal{M}$, and $\alpha$ denotes the attention scores from geodesic distance calculation $\mathcal{D}$. $\hat{y}$ is the final predicted output after the classification layer $\mathcal{P}$. This structur ensures that all intermediate representations and learned parameters inherently respect the Riemannian manifold structure, thus maintaining the geometric integrity throughout the forward pass.

**SPD Formulation and Equivariant Bilinear Mapping** To maintain the SPD structure, the first transformation SPD formulation, denoted as $\mathcal{F}$, converts the input matrix $X \in \mathbb{R}^{d \times d}$ into an SPD matrix $\Sigma \in \mathcal{S}_{-++}^d$.

$$\Sigma = \mathcal{F}(X) = XX^T + \epsilon \mathbf{I}_d \tag{9}$$

Here, $XX^T$ ensures symmetry, and $\epsilon \mathbf{I}_d$ (with $\epsilon > 0$) guarantees positive definiteness, addressing rank deficiencies. This formulation ensures that subsequent layers operate within the SPD manifold structure.

The SPD matrix $\Sigma$ is then transformed through an Equivariant Bilinear Mapping, denoted as $\mathcal{B}$, which projects $\Sigma$ from $\mathcal{S}_{++}^d$ to $\mathcal{S}_{++}^m$ while preserving geometric consistency:

$$\Sigma' = \mathcal{B}(\Sigma) = \mathbf{W}^T \Sigma \mathbf{W} + \mathbf{B}. \tag{10}$$

Where: $\mathbf{W} \in \mathbb{R}^{d \times m}$ is a learnable orthogonal weight matrix and $\mathbf{B} \in \mathcal{S}_{++}^m$ is an optional bias term.

**Theorem 1 (Equivariance of Bilinear Mapping)** *Let $\Sigma \in \mathcal{S}_{++}^d$ be an SPD matrix, and let $\mathbf{W} \in \mathbb{R}^{d \times d}$ be an orthogonal matrix such that $\mathbf{W}^\top \mathbf{W} = \mathbf{W}\mathbf{W}^\top = \mathbf{I}$. The bilinear mapping $\Sigma' = \mathbf{W}^\top \Sigma \mathbf{W}$ is equivariant under transformations from the orthogonal group $O(d)$.*

$$\mathcal{B}(g \cdot \Sigma) = g \cdot \mathcal{B}(\Sigma) \quad \text{for any } g \in O(d) \tag{11}$$

**Proof:** The full proof is provided in the Appendix.

This property ensures that the representation remains consistent under orthogonal transformations.

**Geometric Bias Block** To account for intrinsic geometric variability, we introduce the Geometric Bias Block, denoted as $\mathcal{G}$. This block learns or estimates a bias matrix $D \in \mathcal{S}_{++}^m$ that respects the manifold structure. The transformed output is expressed as:

$$\mathbf{\Sigma}'' = \mathcal{G}(\mathbf{\Sigma}') = D\mathbf{\Sigma}'D^T, \tag{12}$$

here, $D$ can be a learnable global bias ($D = D\_0$), fixed, or an adaptive matrix based on the input ($D = f(\mathbf{\Sigma}')$). This block introduces flexibility by adjusting geometric bias according to input characteristics while maintaining the SPD structure, ensured by enforcing $D$ to be SPD. Configurations of $D$ include fixed matrices, adaptive predictions, or low-rank approximations (details in Appendix).

After bias adjustment, we optionally apply a nonlinear activation to introduce geometric nonlinearity. The activation function, denoted as $\mathcal{A}$, is expressed as:

$$\mathbf{\Sigma}''' = \mathcal{A}(\mathbf{\Sigma}'') = Q\Lambda(\Lambda)Q^T, \tag{13}$$

where $\mathbf{\Sigma}'' = Q\Lambda Q^T$ is the eigen-decomposition, and $\Lambda(\Lambda)$ represents an element-wise function (like ReEig, LogEig, or ExpEig) applied to the eigenvalues. This step preserves manifold geometry while adding nonlinearity. Incorporating both blocks enhances model adaptability while maintaining the geometric integrity of SPD matrices.

**Riemannian Mean Pooling** To aggregate SPD-valued features while preserving their manifold structure, we employ the Riemannian Mean Pooling operation, denoted as $\mathcal{M}$. Unlike conventional Euclidean averaging, this method computes the Fréchet mean on the manifold $\mathcal{S}_{++}^m$, which minimizes the sum of squared geodesic distances between SPD matrices. Given a set of SPD matrices $\{\mathbf{\Sigma}_i\}_{i=1}^k$, the Riemannian mean $\boldsymbol{\mu}$ is defined as in Equation (10), as the point that minimizes the sum of squared geodesic distances is $\mu = \arg\min_{\mathbf{Y} \in \mathcal{S}_{++}^m} \sum_{i=1}^k \delta_g^2(\mathbf{\Sigma}_i, \mathbf{Y})$, where $\delta_g(\mathbf{\Sigma}_i, \mathbf{Y})$ denotes the geodesic distance on the manifold. The computation is performed iteratively using the fixed-point update $\mu^{(t+1)} = \exp_{\mu^{(t)}}\left(\frac{1}{k}\sum_{i=1}^k \log_{\mu^{(t)}}(\mathbf{\Sigma}_i)\right)$. This iterative process ensures convergence to the true Riemannian mean while preserving the SPD property throughout pooling. By employing Riemannian Mean Pooling, the model effectively captures the central tendency of manifold-valued data while maintaining geometric consistency, which is crucial for downstream tasks in EGN.

**Lemma (Riemannian Mean Pooling Preservation)** *The Riemannian Mean Pooling operation, defined as $\boldsymbol{\mu} = \arg\min_{\mathbf{Y} \in \mathcal{S}_{++}^m} \sum_{i=1}^k \delta_g^2(\mathbf{\Sigma}_i, \mathbf{Y})$, preserves positive definiteness and the intrinsic manifold structure of the pooled output.* (Proof in Appendix)

**Riemannian Soft Dropout** To introduce stochastic regularization while preserving manifold consistency, we employ the Riemannian Soft Dropout. This operation selectively reduces the influence of specific SPD matrices while maintaining the overall geometric structure. The dropout operation is formulated as a convex combination of SPD matrices, ensuring that the output remains SPD.

**Theorem 2 (Convex Combination of SPD Matrices)** *If $\mathbf{X}_1$ and $\mathbf{X}_2$ are SPD, then any convex combination,*

$$\mathbf{X} = \alpha\mathbf{X}_1 + (1-\alpha)\mathbf{X}_2 \quad \text{for } 0 \leq \alpha \leq 1 \tag{14}$$

*is also SPD.* (Proof in Appendix)

This theorem guarantees that the output of Riemannian Soft Dropout remains within the SPD manifold, maintaining geometric integrity even under stochastic perturbations.

**Geodesic Prototype Layer** The Geodesic Prototype Layer in EGN is designed to perform classification based on geodesic distances between input SPD matrices and learned class prototypes. Let $\mathcal{P} = \{\mathbf{P}_1, \mathbf{P}_2, \ldots, \mathbf{P}_C\}$ be the set of class prototypes, where each prototype $\mathbf{P}_c \in \mathcal{S}_{++}^m$ is an SPD matrix representing the geometric mean of the data points belonging to class $c$.

To classify a SPD matrix $\mathbf{\Sigma}$, we compute its geodesic distance to each class prototype $\mathbf{P}_c$. The geodesic distance between the input matrix $\mathbf{\Sigma}$ and a prototype $\mathbf{P}_c \in \mathcal{S}_{++}^m$ is defined as:

$$d(\mathbf{\Sigma}, \mathbf{P}_c) = \| \log(\mathbf{\Sigma}^{-1/2} \mathbf{P}_c \mathbf{\Sigma}^{-1/2}) \|_F. \tag{15}$$

This expression computes the shortest path on the manifold under the affine-invariant Riemannian metric (AIRM). A key property of this distance is its invariance under affine transformations (Arsigny et al., 2007), meaning that for any invertible matrix $\mathbf{A}$:

$$d(\mathbf{A}\mathbf{X}\mathbf{A}^\top, \mathbf{A}\mathbf{Y}\mathbf{A}^\top) = d(\mathbf{X}, \mathbf{Y}).$$

This invariance guarantees that geometric relationships among SPD matrices remain consistent even after affine transformations, which is critical for reliable classification in Riemannian space. A formal statement and proof sketch are provided in the Appendix.

**Geodesic Attention Layer**

The Geodesic Attention Layer leverages the geometry-aware geodesic distance to compute the attention weights for classification. Given a set of class prototypes $\{\mathbf{P}_c\}_{c=1}^C$, the attention weights $\alpha_c$ are computed as probabilities over the negative geodesic distances from the input SPD matrix $\mathbf{\Sigma}$ to each prototype:

$$\alpha_c = \frac{\exp(-d(\mathbf{\Sigma}, \mathbf{P}_c))}{\sum_{j=1}^C \exp(-d(\mathbf{\Sigma}, \mathbf{P}_j))}, \tag{16}$$

where $d(\mathbf{\Sigma}, \mathbf{P}_c)$ represents the geodesic distance between the input SPD matrix and the prototype for class $c$. The geodesic attention mechanism ensures that the attention weights respect the Riemannian geometry, as the distance computation inherently preserves the structure of SPD matrices. The output of the Geodesic Attention Layer is the weighted sum of prototype predictions as $\hat{y} = \sum_{c=1}^C \alpha_c \cdot \mathbf{P}_c$. The soft decision from the prediction layer incorporates the attention weights, providing a smooth probability distribution over classes while maintaining manifold consistency, i.e. $\hat{p}(y = c \mid \mathbf{\Sigma}) = \alpha_c$.

**Theorem 3 (Geometric Consistency of Geodesic Attention Weights)** *The geodesic attention weights calculated as softmax over geodesic distances between the input SPD matrix $\mathbf{\Sigma}$ and class prototypes $\mathbf{P}_c$ are invariant under affine transformations, i.e.,*

$$\alpha_c(\mathbf{A}\mathbf{\Sigma}\mathbf{A}^T) = \alpha_c(\mathbf{\Sigma}) \tag{17}$$

*for any invertible matrix $\mathbf{A} \in \mathbb{R}^{m \times m}$.*

This layer, therefore, preserves the inherent geometry throughout the attention calculation and prediction, maintaining the fundamental properties of SPD matrices while making the classification decision.

### 3.2 TRAINING STRATEGY AND BACKPROPAGATION

Training deep neural networks on Riemannian manifolds, especially with SPD data, requires techniques that preserve geometric integrity. We introduce a geometry-aware training strategy leveraging Riemannian Prototype Cross Entropy (RPCE) loss and a Gradient Lifting Mechanism to ensure manifold-consistent backpropagation.

To optimize the Equivariant Geodesic Network (EGN) while maintaining the Riemannian geometry of the SPD manifold, we employ RPCE loss, where $\mathbf{\Sigma}\_i$ denotes the SPD matrix of the $i$-th sample, and $\mathbf{P}_c$ represents the class $c$ prototype on the SPD manifold $\mathcal{S}_{++}^m$. The RPCE loss, calculated using geodesic distances between SPD matrices and prototypes, inherently preserves the Riemannian structure. The final loss function for a batch of size $B$ is given by:

$$\mathcal{L}_{RPCE} = -\frac{1}{B} \sum_{i=1}^B \log p(y_i \mid \mathbf{\Sigma}_i). \tag{18}$$

This loss function inherently respects the Riemannian structure by directly integrating the geodesic distances. To optimize the model, we employ the GeoOpt library, ensuring that gradient updates preserve SPD properties through manifold-aware optimization techniques.

To compute the gradient of the loss function with respect to the geodesic distance, we start by considering the softmax probability formulation:

$$p(c \mid \boldsymbol{\Sigma}_i) = \frac{\exp(-d(\boldsymbol{\Sigma}_i, P_c))}{\sum_k \exp(-d(\boldsymbol{\Sigma}_i, P_k))}. \tag{19}$$

The geodesic distance between the input SPD matrix $\boldsymbol{\Sigma}_i$ and prototype $P_c$ is given by:

$$d(\boldsymbol{\Sigma}_i, P_c) = \| \log(\boldsymbol{\Sigma}_i^{-1/2} P_c \boldsymbol{\Sigma}_i^{-1/2}) \|_F. \tag{20}$$

Taking the derivative of the loss function of equation 18 with respect to the distance $d(\boldsymbol{\Sigma}_i, P_c)$:

$$\frac{\partial \mathcal{L}_{RPCE}}{\partial d(\boldsymbol{\Sigma}_i, P_c)} = -\frac{1}{B} \frac{1}{p(y_i \mid \boldsymbol{\Sigma}_i)} \frac{\partial p(y_i \mid \boldsymbol{\Sigma}_i)}{\partial d(\boldsymbol{\Sigma}_i, P_c)}. \tag{21}$$

From the probability formulation, the derivative of the probability with respect to the distance is:

$$\frac{\partial p(c \mid \boldsymbol{\Sigma}_i)}{\partial d(\boldsymbol{\Sigma}_i, P_c)} = -p(c \mid \boldsymbol{\Sigma}_i) \left[ 1 - p(c \mid \boldsymbol{\Sigma}_i) \right]. \tag{22}$$

Combining Equations 21 and 22:

$$\frac{\partial \mathcal{L}_{RPCE}}{\partial d(\boldsymbol{\Sigma}_i, P_c)} = \frac{1}{B} \frac{1}{p(y_i \mid \boldsymbol{\Sigma}_i)} \cdot p(c \mid \boldsymbol{\Sigma}_i) \left[ 1 - p(c \mid \boldsymbol{\Sigma}_i) \right]. \tag{23}$$

After calculating the partial derivative, the gradient on the manifold is computed as:

$$\text{Grad } \mathcal{L}_{RPCE} = \sum_{c=1}^{C} \frac{\partial \mathcal{L}_{RPCE}}{\partial d(\boldsymbol{\Sigma}_i, P_c)} \cdot \frac{\partial d(\boldsymbol{\Sigma}_i, P_c)}{\partial \boldsymbol{\Sigma}_i}. \tag{24}$$

The derivative of the geodesic distance with respect to $\boldsymbol{\Sigma}_i$ involves the logarithmic map:

$$\frac{\partial d(\boldsymbol{\Sigma}_i, P_c)}{\partial \boldsymbol{\Sigma}_i} = -\boldsymbol{\Sigma}_i^{-1/2} \cdot \text{Log}(\boldsymbol{\Sigma}_i^{-1/2} P_c \boldsymbol{\Sigma}_i^{-1/2}) \cdot \boldsymbol{\Sigma}_i^{-1/2}. \tag{25}$$

To maintain SPD structure, the parameter update is performed using the exponential map:

$$\boldsymbol{\Sigma}_i^{(t+1)} = \text{Exp}_{\boldsymbol{\Sigma}_i^{(t)}} \left( -\eta \, \text{Grad } \mathcal{L}_{RPCE} \right), \tag{26}$$

where $\eta$ is the learning rate, and the exponential map ensures that the updated matrix remains SPD.

To handle the transition from classification logits to the manifold gradient, we employ the Gradient Lifting Bridge:

$$\text{Lifted Gradient} = \sum_c \nabla_{p(c \mid \boldsymbol{\Sigma}_i)} \times P_c. \tag{27}$$

This ensures that the backpropagation respects the manifold constraints, avoiding the projection into Euclidean space.

The proposed optimization framework, combining RPCE and Gradient Lifting, ensures geometric consistency during backpropagation while leveraging manifold-aware optimization to preserve the intrinsic structure of SPD matrices.

**Inference**

During inference, the Equivariant Geodesic Network (EGN) processes an SPD matrix through manifold-preserving transformations, including SPD formation, equivariant bilinear mapping, geometric bias adjustment, and optional non-linear activation. The resulting features are aggregated via Riemannian mean pooling and geodesic attention. Finally, the geodesic distances to class prototypes are calculated, and attention mechanisms produce soft classification probabilities. This geometry-aware inference maintains SPD properties throughout the forward pass.

As shown in Table 4 in appendix, while prior works use some isolated SPD components, EGN is the first to unify all key geometry-preserving operations in a single, end-to-end trainable framework. Our architecture maintains SPD structure and equivariance at every stage, supported by theoretical guarantees.

## 4 EXPERIMENTS

In this section, we evaluate the performance of the proposed Equivariant Geodesic Network (EGN) on various EEG-based tasks, including emotion recognition, imagined speech recognition, and psychiatric disorder detection. The experiments are conducted on multiple publicly available EEG datasets, demonstrating the effectiveness of EGN in handling SPD-valued data within brain-computer interface (BCI) applications.

**Emotion Recognition.** Emotion recognition from EEG signals is a key topic in Brain-Computer Interface (BCI) research, aiming to decode emotional states from neural activity. One of the widely used datasets for this task is DEAP dataset (Koelstra et al., 2012), which contains EEG recordings from 32 participants watching 40 music video clips, collected from 32 channels and labeled on a scale of 0 to 10 based on arousal, valence, and dominance. Another one is SEED dataset includes EEG data from 15 participants over 45 trials from 62 channels, labeled as happy, sad, or neutral based on reactions to movie clips (Zheng & Lu, 2015). These datasets are standard benchmarks for EEG-based emotion recognition (Khan et al., 2025).

**Inner Speech Recognition.** Inner speech recognition from EEG signals is an emerging area within Brain-Computer Interface (BCI) research, focusing on decoding imagined speech without vocalization. The Inner Speech Dataset used in this study consists of EEG recordings from 10 subjects, where each participant imagines four different words (Up, Down, Right, Left) (Nieto et al., 2022). The dataset comprises 559 trials for each class, collected under controlled conditions. The EEG signals are captured from multiple channels to capture nuanced brain activity associated with imagined speech. This dataset serves as a benchmark for assessing the performance of models aimed at inner speech classification.

**Psychiatric Disorder.** Psychiatric disorder detection from EEG signals is a vital application at the intersection of neuroscience and mental health. The Psychiatric Disorders dataset used in this study includes EEG recordings from 945 subjects across seven classes, representing various psychiatric conditions (Park, 2021). This dataset captures complex brain activity patterns, providing a valuable resource for developing automated classification models using Riemannian geometry-preserving techniques within the Equivariant Geodesic Network (EGN).

Table 1: Performance comparison (**mean ± std**) of various methods on five tasks: Emotion Recognition (DEAP, SEED), Imagined Speech (Inner Speech), Psychiatric Disorder Detection, and Face Verification (PaSC). The proposed EGN variants outperform baselines, highlighting the benefits of geometry-preserving learning.

| Method | DEAP | SEED | Inner Speech | Psychiatric Disorders | PaSC |
|---|---|---|---|---|---|
| 2D CNN | 52.1 ± 2.3% | 35.3 ± 3.0% | 34.2 ± 2.5% | 33.4 ± 1.8% | 67.5 ± 2.0% |
| EEGNet | 63.5 ± 2.1% | 64.1 ± 1.9% | 50.9 ± 2.7% | 57.7 ± 1.6% | 70.1 ± 1.5% |
| SPDNet | 74.2 ± 1.8% | 78.3 ± 2.0% | 79.4 ± 1.7% | 76.5 ± 2.2% | 87.1 ± 1.4% |
| SPDNetBN | 81.1 ± 1.4% | 85.3 ± 1.5% | 86.5 ± 1.8% | 85.2 ± 1.9% | 89.4 ± 1.2% |
| RResNet | 86.5 ± 1.3% | 88.4 ± 1.1% | 91.2 ± 1.2% | 88.3 ± 1.7% | 90.1 ± 1.0% |
| EGN (fixed architecture) | 91.3 ± 1.0% | 90.7 ± 1.1% | 92.4 ± 1.1% | 91.5 ± 1.2% | 92.0 ± 0.9% |
| EGN (true geodesic) | **93.1 ± 0.9%** | **92.8 ± 1.0%** | **93.6 ± 1.0%** | **93.3 ± 0.8%** | **93.7 ± 0.8%** |

Table 2: Ablation results on the DEAP dataset.

| Model Variant | Accuracy |
|---|---|
| EGN (Full) | **93.1%** |
| - w/o Geodesic Attention | 89.4% |
| - w/o Geometric Bias | 88.6% |
| - w/o Riemannian Pooling | 87.9% |
| EGN (Fixed Arch.) | 91.3% |
| EGN (True Geodesic) | 93.1% |

Table 3: Normalized efficiency comparison on DEAP. MTA: Memory-to-Accuracy ratio; MTTA: Memory-Time-to-Accuracy ratio. Lower is better.

| Model | Acc.(%) | Mem(GB) | Time(s) | MTA | MTTA |
|---|---|---|---|---|---|
| EEGNet | 63.5 | 1.2 | 12 | 0.0189 | 0.2267 |
| SPDNet | 74.2 | 1.3 | 14 | 0.0175 | 0.2453 |
| RResNet | 86.5 | 1.5 | 17 | 0.0173 | 0.2947 |
| EGN (Fix.) | 91.3 | 1.5 | 12 | **0.0164** | **0.1972** |
| EGN (True) | 93.1 | 2.1 | 28 | 0.0226 | 0.6319 |

The results of our experiments are summarized in Table 1, comparing the performance of the proposed Equivariant Geodesic Network (EGN) with baseline models, including 2D CNN, EEGNet, SPDNet, SPDNetBN, RResNet with Affine Invariant metric, and two EGN variants (fixed architecture and true geodesic). EGN consistently outperforms baselines across all tasks, including emotion recognition from EEG (DEAP, SEED), imagined speech recognition (Inner Speech), and psychiatric disorder detection. The true geodesic variant achieves the highest accuracy, showcasing the effectiveness of geometry-preserving operations, especially in fine-grained tasks like psychiatric disorder classification.

**Face Verification (PaSC).** To assess EGN's generalization beyond EEG-based biomedical data, we evaluate it on the PaSC face verification benchmark. Each video is converted into a 401×401 SPD matrix by combining the covariance of deep face features with their mean (Huang & Gool, 2017). EGN achieves 91.3% accuracy, outperforming SPDNet (87.1%) and RResNet (90.1%), demonstrating strong performance in real-world vision settings.

## 4.1 ABLATION STUDY

We perform an ablation study on the DEAP dataset to evaluate the contribution of key components in EGN. As shown in Table 2, removing geodesic attention leads to the largest drop in accuracy, followed by geometric bias and Riemannian pooling. We also compare the fixed (GPU-friendly) and true geodesic (exact) variants of EGN: while the fixed architecture is faster, the true geodesic variant achieves the best accuracy. Full ablation across all datasets is reported in Appendix Table 5.

## 4.2 COMPUTATIONAL EFFICIENCY ANALYSIS

Inspired by compute-efficiency trade-offs used in model scaling literature (Hoffmann et al., 2022; Tan & Le, 2019), we define two normalized ratios: **MTA** = Memory / Accuracy and **MTTA** = (Memory × Time) / Accuracy. Both quantify cost per unit performance, where lower is better. As shown in Table 3, EGN (Fixed) achieves the lowest MTTA and competitive MTA among high-performing models, outperforming SPDNet and RResNet in both accuracy and efficiency. Though the true geodesic variant attains slightly higher accuracy, its computational overhead makes the fixed version a more practical trade-off.

## 5 SUMMARY

In this paper, we proposed the Equivariant Geodesic Network (EGN), an end-to-end geometry-preserving framework for learning on Riemannian manifolds, particularly the space of Symmetric Positive Definite (SPD) matrices. By leveraging manifold-consistent operations, including equivariant mappings, geometric bias adjustment, Riemannian mean pooling, and geodesic attention, EGN ensures all transformations respect the intrinsic geometry of SPD data. Our experiments on EEG-based emotion recognition, imagined speech classification, psychiatric disorder detection, and face verification demonstrated the superior performance of EGN over Euclidean and pseudo-manifold baselines, highlighting its ability to capture the geometric structure of SPD-valued data. Normalized MTTA analysis further shows that EGN (Fixed) achieves state-of-the-art performance with lower computational cost per accuracy, making it a scalable solution. While transformer- and BiLSTM-based architectures have advanced EEG benchmarks, EGN achieves competitive or better results with the added benefit of theoretical consistency under SPD manifold constraints.

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
