# OpenReview forum: "Equivariant Geodesic Networks: Geometry Preserving Learning on Riemannian Manifolds"
_ICLR.cc/2026/Conference — ICLR 2026 Conference Withdrawn Submission_

### Official Review · Reviewer_eLPk · 2025-10-20

**Soundness:** 2
**Presentation:** 1
**Contribution:** 1
**Rating:** 2
**Confidence:** 4

**Summary:**

The paper proposes an end-to-end Equivariant Geodesic Network (EGN) for learning on the SPD manifold. The pipeline includes SPD construction from raw inputs, an equivariant bilinear mapping, a “geometric bias” block, spectral-domain activations, Riemannian mean pooling, and a geodesic attention classifier with prototype distances under an affine-invariant metric. The authors claim geometry-preserving backpropagation and introduce “Riemannian prototype cross-entropy.” Experiments cover several EEG benchmarks and a face verification task, with custom efficiency metrics that combine accuracy, memory, and time.

**Strengths:**

* The paper designs a fully geometry-aware pipeline on SPD, from input construction through to classification.
* Its emphasis on equivariance and affine-invariant geodesic distances is conceptually sound for SPD.
* The authors also attempt to assess efficiency and include a few ablations.

**Weaknesses:**

* Notation is inconsistent across sections, with indices and matrix dimensions changing or left implicit. Several symbols are overloaded or undefined at first use.
* Cross references among equations and sections are occasionally incorrect or ambiguous, which makes it hard to follow the derivations.
* The output semantics in the “geodesic attention” stage are unclear. It appears to produce both a manifold-valued combination of class prototypes and class probabilities. The mapping between these quantities and the final loss is not fully specified.
* Key algorithmic components are described at a high level. There is no clear block diagram or concise pseudocode that lists inputs, outputs, and invariants of each module.
* Many modules resemble standard SPD operations used in prior Riemannian neural networks: $XX ^\top+\epsilon I$ SPD construction, congruence transforms $W ^\top \Sigma W$, spectral filtering or ReEig-like activations, Karcher or Riemannian mean pooling, and prototype classification under AIRM. The paper does not convincingly isolate what is fundamentally new from the reassembly of known components.
* The “geometric bias” block is insufficiently motivated. Its parameterization, impact on equivariance, and positive-definiteness guarantees are not clearly justified or analyzed.
* Baselines do not sufficiently cover the current manifold learning landscape. Several strong SPD or hyperbolic baselines are missing.

**Questions:**

* Please unify notation, explicitly list tensor and matrix shapes per layer, and add a compact algorithm box or pseudocode for the full forward and backward passes.
* Clarify the semantics of geodesic attention. If the model outputs a manifold-valued combination of prototypes and a probability vector, define the exact mapping to the loss and explain why that is consistent.
* Include statistical tests where appropriate.
* Strengthen baselines. Add representative SPD manifold networks, as well as hyperbolic or other non-Euclidean baselines.
* Expand ablations. Isolate the contribution of each proposed module on multiple datasets. Include sensitivity to projection ranks, regularization, and spectral activation choices.

---

### Official Review · Reviewer_Z27y · 2025-10-30

**Soundness:** 1
**Presentation:** 2
**Contribution:** 2
**Rating:** 2
**Confidence:** 4

**Summary:**

The present paper introduces a novel architecture for neural networks operating on SPD matrices. It is distinguished from previous architectures by performing all options on the manifold of SPD matrices. To this end, various layers are introduced to construct this network. The new architecture is tested on a range of datasets, primarily EEG data, and exhibits improved performance while using fewer resources.

**Strengths:**

- The authors introduce a novel architecture that comes with a set of building blocks potentially interesting in a larger context.
- The introduced architecture exhibits improved performance on the investigated datasets. Notably this holds true both in terms of accuracy and efficiency.

**Weaknesses:**

*Correctness:*
This is maybe the most important weakness of the paper: there are simply a range of factual errors. For example, theorem 1 is wrong. The provided proof shows that the matrix is conjugated by a *different* matrix! Plus, what does equivariance even mean when $d\neq m$ ? It is not the same group operating. Furthermore, the proof of theorem 3 shows that $\alpha$ is only invariant if $P_c$  is also transformed, which is not part of the statement in (17)! The authors need to be more careful what they can actually prove.

*Clarity:*
This is the second major issue with the present version of the paper is the clarity of the exposition. The method is unfortunately not clearly described from my point of view. Meaning that with the paper in hand, I would not be able to reproduce the method. Multiple of the building blocks are not described in enough detail, if at all. See questions below for a few more concrete pointers. My recommendation would be to brush up the text and include all necessary details (some maybe in the appendix), and, at the same time, provide pseudo-code for the method. I believe this would make the reader's experience a lot better and increase the impact of the paper. Also, the description of the experiments is incomplete, e.g. I could not find information on how train/test splits where performed. Again, the author's should provide all information necessary to reproduce the experiments in the paper and appendix.

*Overselling:*
I feel that the paper oversells its contribution in two ways: First, in the beginning it talks a lot about general Riemannian manifold, when in fact it only works with the manifold with SPD matrices with the AIRM. Furthermore, the preservation of SPDness is stressed at many points without providing evidence why it is beneficial at this specific point.

In summary, these weaknesses lead me to recommend to reject the paper for now. To me, it feels like the author's rushed to submission and the paper needs more work to be ready for the spotlight even though the method might be interesting.

**Questions:**

Methodology:
  - Why is the bias block called a bias?
  - Riemannian Mean Pooling: Which matrices are averaged?
  - Soft Dropout: How does it actually work? Which matrices are convexly combined?
  - Prototype Layer + Attention: To me, this is not attention, this is just a softmax classifier based on the distance to precomputed means of the classes. Attention would look at the relationship between matrices in a sequence?
  - Training Strategy: What is going on?  Here I am admittedly lost. Why is the sample updated? How is this used to update the weights of the network?

Experiments:
   - What is the fixed version of the EGN?
   - What is the training protocol? (e.g. how is the training/test split done, this seems pretty critical on such small datasets)
   - What are the details of the PaSC experiment? What is the performance of SOTA methods not using SPD matrices?
   - Efficiency: I assume this is for inference?

Appendix:
 - Please move the appendix into the main document.
 - I would recommend to include the related work in to the main text. Position a new method in context of existing work is crucially important.

Minor things/suggestions from my side:
  - l. 37: "projecting can distort" -> I would recommend to be more specific
  - l. 46: S_++
  - Background: No need for subsections
  - l. 97: g_p(v,w) = <v,w>\_p is meaningless without explanation
  - l. 100: I find this to be a unusual reference for Riemannian geometry -> I would recommend standard textbook
  - l. 106: please improve typesetting
  - Methodology: I found preservation of SPD to be overstressed to a point where it hinders readability

---

### Official Review · Reviewer_2kHo · 2025-11-01

**Soundness:** 2
**Presentation:** 2
**Contribution:** 2
**Rating:** 2
**Confidence:** 5

**Summary:**

This paper proposes Equivariant Geodesic Networks (EGN), an end-to-end neural architecture operating on the manifold of Symmetric Positive Definite (SPD) matrices. The model integrates SPD-forming layers, equivariant bilinear mapping, a geometric bias block, eigenvalue-domain nonlinearities, Riemannian mean pooling, and geodesic prototype attention classification. Experiments on EEG emotion recognition, imagined speech, psychiatric disorder detection, and PaSC face verification show competitive results compared to SPDNet and EEGNet.

**Strengths:**

The architecture maintains SPD structure throughout the network, ensuring geometric consistency under the affine-invariant Riemannian metric.

**Weaknesses:**

## Theoretically
- Several results, such as the lemma on Riemannian mean pooling and Theorems 2–3, restate well-known or trivial facts. There is no real value in presenting them as new theorems, as these properties are standard results in the SPD literature.
- The geometric bias block is essentially a gyroaddition bias in SPD space, a concept that has already been formally analyzed in prior studies [b]. However, the paper provides no citation and discussion.
- The proposed geodesic attention is a simplified version of MAtt [e], yet this work is neither cited nor experimentally compared, giving an incomplete view of the research landscape.
- The proposed layer design relies heavily on SPDNet, while overlooking several recent advances that already developed manifold-consistent fully connected, convolutional, and attention layers [a, c, d]. These methods have demonstrated strong performance for SPD learning.
- The claimed “equivariance” is theoretically weak compared to ManifoldNet [i]. In ManifoldNet, equivariance is defined with respect to the isometry group, whereas in this submission, it is reduced to O(n), a subset of GL(n) for AIM. Besides, the equi- and in-variance did not work under dimension change.


## Empirically
- Many key baselines are missing. It is unusual that none of the manifold-attention models are included in the experiments, such as [a, e–f]. Moreover, several recent SPD-based EEG methods (e.g., TSMNet) are omitted, which significantly weakens the empirical evidence for novelty and superiority.

-[a] Neural Networks on Symmetric Spaces of Noncompact Type
-[b] The Gyro-Structure of Some Matrix Manifolds
-[c] Building Neural Networks on Matrix Manifolds: A Gyrovector Space Approach
-[d] MATRIX MANIFOLD NEURAL NETWORKS++
-[e] MAtt: A Manifold Attention Network for EEG Decoding
-[f] Robust Hyperbolic Learning with Curvature-Aware Optimization
-[g] A Grassmannian Manifold Self-Attention Network for Signal Classification
-[h] A Correlation Manifold Self-Attention Network for EEG Decoding
-[i] ManifoldNet: A Deep Neural Network for Manifold-Valued Data With Applications

**Questions:**

see wk

---

### Official Review · Reviewer_GGJo · 2025-11-04

**Soundness:** 2
**Presentation:** 2
**Contribution:** 1
**Rating:** 2
**Confidence:** 5

**Summary:**

The underlying space of SPD matrices is actually a non-Euclidean Riemannian manifold, i.e., SPD manifold, making the existing Euclidean-based representation learning methods and computations can not be applied directly. In this work, the authors propose an Equivariant Geodesic Networks (EGN) on the Riemannian manifolds that incorporates manifold-consistent operations, including equivariant mappings, adaptive geometric bias, and structured low-rank updates. Unlike existing methods that either flatten or project SPD data points onto Euclidean space, EGN directly learns on the manifold, preserving geometric consistency in an end-to-end manner. Extensive experiments on several EEG-based tasks confirm the effectiveness of the proposed model.

**Strengths:**

1. This paper is easy to follow.

  2. The geodesic attention-based signal classification seems a bit novel, as it operates directly on the SPD manifolds. In contrast, most of the existing SPD networks perform classification in the Euclidean space using manifold-to-Euclidean embedding mapping (e.g., LogMap + FC + Softmax).  Therefore, the proposed classification mechanism is qualified to produce higher accuracy.

  3. Experimental validity.

**Weaknesses:**

1. The  novelty of this method is limited. The reviewer is familiar with the field of Riemannian manifold learning, the designed manifold-valued operations, like equivariant mappings, adaptive geometric bias, Riemannian mean pooling, already have been studied in a number of previous works, such as SPDNet [Huang, et al, AAAI, 2017], SPDNetBN [Brooks, et al, NeurIPS, 2019], U-SPDNet [Wang et al, Neural Networks, 2023], Matt [Pan, et al, NeurIPS, 2022]. This diminishes the originality of this article.

2. There may be problems with the proof of Eq. 11. In the Appendix, to make $\mathcal{B}(g\cdot \Sigma)$ = $\tilde{g}\mathcal{B}(\Sigma)\tilde{g}^T$ hold ($\tilde{g}=W^TgW$), $g$ should also be a symmetric matrix, such that $g^T=g$. However, in my opinion, such a matrix is ​​very special and not well-suited to the computational scenario. In other words, $g^2=I$, which means that the eigenvalues of g  can only be 1 or -1.

3. Although the including of the optional bias term $B$ (Eq. 10) will not destroy the Riemannian geometry of the data points, it does not "walk along the geodesic". In other words, the resulting data point will deviate from the actual Riemannian barycenter. Besides, Eq. 14 has the same issue, as it  performs weighted combination using Euclidean paradigm. The reviewer suggests the authors re-design Eq. (14) using the Riemannian metric-based weighted Fréchet mean (geodesic interpolation), which is an intrinsic method.

4.  In the experiments, the details are lacking, i.e., the preprocessing pipelines, number of subjects used, data splits, hyperparameter tuning, and statistical tests are not reported.

5. The Reproducibility statement is missing, violating the ICLR submission requirements.

6. The authors should report statistical significance and clarify whether the gain arises from manifold-valued operations or architectural size.

7. Eq. (25) derivative of geodesic distance lacks justification for matrix logarithm differentiation. This requires citing Pennec (2006) or Sra (2012).

8. This papers lacks some necessary citations for the previous proposed operations in Section 3.

Minor aspects:

1. Symmetric positive definite (SPD) matrices should be given their full name and abbreviation upon their first appearance; subsequent appearances can use the abbreviation SPD.

2. The citation of a reference cannot jump to the bibliography section. There is a problem with the settings.

3. Line 46, page 1; line 198, page 4, S_++^n, should in a correct mathtype form.

4. Line 225, page 5, D_0, should also in a correct mathtype form.

5. Please check Eq. 12 and Eq. 13. If they are correct, the positions of $\mathcal{G}$ and $\mathcal{A}$ in the formulation of line 188 should be reversed.

**Questions:**

1. The motivation expressed in the paper is not clear or specific enough. The authors repeatedly emphasize that "existing methods only partially respect manifold properties," but do not clearly explain what the substantive limitations of existing methods (such as SPDNet, SPDNetBN, RResNet) are. For instance, whether they are due to approximation errors in tangent space mapping, lack of equivariance, or instability of Riemannian backpropagation.

2. Could the authors clearly specify which parts of EGN are theoretically new rather than architectural combinations?

3. What is the basis for differentiating the matrix logarithm in Eq. (25)?

4. The paper introduces the Riemannian Prototype Cross-Entropy (RPCE) loss. How does RPCE differ mathematically from a standard cross-entropy applied to geodesic distances or softmax over tangent-space embeddings?

5. Could the authors provide exact experimental protocols (data splits, trial counts per class, normalization, etc.) and confirm whether all datasets are subject-independent?

6. Could the authors clarify how to construct SPD matrices from facial features and whether EGN truly outperforms Euclidean CNN baselines.

7. This paper lacks some visualization results, like t-SNE, to confirm that the proposed EGN can indeed preserve the manifold structure.

8. This submission seems to lack the required Ethics Statement and Reproducibility Statement sections.

---

### Note · Authors · 2026-01-10

I have read and agree with the venue's withdrawal policy on behalf of myself and my co-authors.